# A Logistic Approach for Kinetics of Isothermal Pyrolysis of Cellulose

Jorge López-Beceiro, Ana María Díaz-Díaz , Ana Álvarez-García, Javier Tarrío-Saavedra, Salvador Naya and Ramón Artiaga *

Escola Politécnica Superior, University of A Coruña, Avda. Mendizábal s/n, 15403 Ferrol, Spain; jorge.lopez.beceiro@udc.es (J.L.-B.); ana.ddiaz@udc.es (A.M.D.-D.); ana.alvarez1@udc.es (A.Á.-G.); javier.tarrio@udc.es (J.T.-S.); salvador.naya@udc.es (S.N.)
* Correspondence: ramon.artiaga@udc.es; Tel.: +34-881-013-202

**Abstract:** A kinetic model is proposed to fit isothermal thermogravimetric data obtained from cellulose in an inert atmosphere at different temperatures. The method used here to evaluate the model involves two steps: (1) fitting of single time-derivative thermogravimetric curves (DTG) obtained at different temperatures versus time, and (2) fitting of the rate parameter values obtained at different temperatures versus temperature. The first step makes use of derivative of logistic functions. For the second step, the dependence of the rate factor on temperature is evaluated. That separation of the curve fitting from the analysis of the rate factor resulted to be very flexible since it proved to work for previous crystallization studies and now for thermal degradation of cellulose.

**Keywords:** kinetics; isothermal; cellulose; pyrolysis; degradation

## 1. Introduction

Cellulosic biomass is the most abundant bioresource produced on earth and cellulose is one of the common compounds in ordinary life [1]. Pulp is used in a vast number of different end-uses, but the main applications areas are tissue, board, printing, and specialty paper manufacturing as well as in textiles [2]. Cellulosic materials are generally considered to have a low static propensity, and hence fabrics such as cotton are the preferred materials for use in applications where electrostatic discharges must be avoided [3]. There has been increased social awareness in promotion of environmentally friendly materials, paving the way for further cellulose-based research [4].

Thermal degradation of cellulose has been of interest for a longtime. A number of studies were focused on thermal degradation kinetics of cellulose, hemicellulose, and lignin [5–14]. As described for other materials, in open system, the initial water content should not affect the decomposition temperature if all water had evaporated from samples prior to reaching the decomposition temperature [15]. However, the presence of hemicellulose, lignin, and other impurities or additives would radically affect the thermal behavior of cellulose and chemical reactions occurring during pyrolysis [16]. Thermogravimetry (TG) was even tested as an alternative method for the characterization of archaeological wood [17]. It is one of the most frequently used techniques to evaluate thermal stability and degradation kinetics of polymeric materials. A system based on radiative heat transfer allows for the continuous monitoring of the sample temperature and weight, while the heating rates are much faster than those usually encountered in standard thermogravimetric analyses [18,19]. Pyrolysis of cellulose was studied from thermogravimetric data through several kinetic models and procedures. In general, an Arrhenius dependence of the rate factor on temperature is assumed and activation energies in the 50 to 300 kJ mol$^{-1}$ range were reported [20–22]. Most kinetic methods can be classified into the model-fitting or into the model-free groups. Advantages and drawbacks of the methods were commented in a previous work of the authors where a model was proposed to account for isothermal

and non-isothermal data [11]. While a number of studies provide reasonably good matching of measured and estimated conversions in a relatively broad range of conversions or temperatures, the author's aim with this work was to improve and extend the matching to all reliable experimental data. Some works demonstrated that some differences of temperature may exist into the sample when using non-isothermal conditions [20,23]. Those differences can be relevant in kinetic studies. Thus, the present study is exclusively based on isothermal experimental data.

The aim of the work is to obtain an accurate mathematical description of the degradation rate with respect to the temperature. For that, using a similar approach to that described in a recent crystallization kinetics study [24], optimal fittings of individual isothermal TG curves are performed. The method for the fittings was previously described [11,25,26]. In this work the average squared error (ASE) is independently minimized in the different fittings [27]. Thus, values of the rate parameter were obtained at different temperatures, and then, the temperature dependence of the rate parameter was evaluated. In addition, it is sought that all parameters used in the model have a clear physical meaning. In fact, one of the goals of the parametric regression models against the nonparametric model fitting alternatives is the interpretability of their parameters from a chemical-physical point of view. The regression models used for the fittings in this work are parametric and nonlinear, based on the logistic and exponential functions [28]. The model-fitting approach used in this work is new and consists of using a different function for fitting the rate parameter than for the isothermal time derivative thermogravimetric (DTG) curves. Moreover, no reports have been found on multi-temperature fits as good as those presented here. In the end, this work allows estimating, with relative accuracy and precision, the rate of cellulose degradation at a given temperature.

## 2. Materials and Methods

The material used is a micro-granular high-purity cellulose powder for partition chromatography purchased from Sigma-Aldrich (St. Louis, MO, USA). It has a density of 0.6 $gcm^{-3}$ and was certified as a 0.0% residue on ignition, 3 ppm of Iron and 0.1 ppm of Copper.

Thermogravimetric experiments were performed in a TA Instruments SDT 2960 device. That instrument is a simultaneous TG-differential scanning calorimetry (DSC) device in which the sample temperature is measured through a thermocouple, which is located in contact with the sample platform.

A linearly heating ramp experiment at 20 °C/min was conducted with a zinc sample for temperature calibration.

All experiments were conducted with a 100 mL $min^{-1}$ nitrogen purge using open alumina crucibles. In all cases, sample mass was in the range from 8.5 to 9.5 mg.

The experimental setup consisted of a 30 °C $min^{-1}$ heating ramp up to the isothermal temperature, followed by an isothermal step. Duration of the isothermal step was programmed in excess. Experiments were manually stopped when most of the degradation process seemed to be completed.

Temperatures for isothermal tests were chosen from a preliminary ramp test at 10 °C/min. The temperatures were chosen along the raising part of the DTG peak in the range from 279 to 301 °C, as displayed in Figure 1. Using a wide range of temperatures would result in very different mass losses at each temperature, which can be explained through competitive degradation reactions of cellulose [19]. However, this work aims to analyze a single process and, thus, a narrow range of temperatures is chosen.

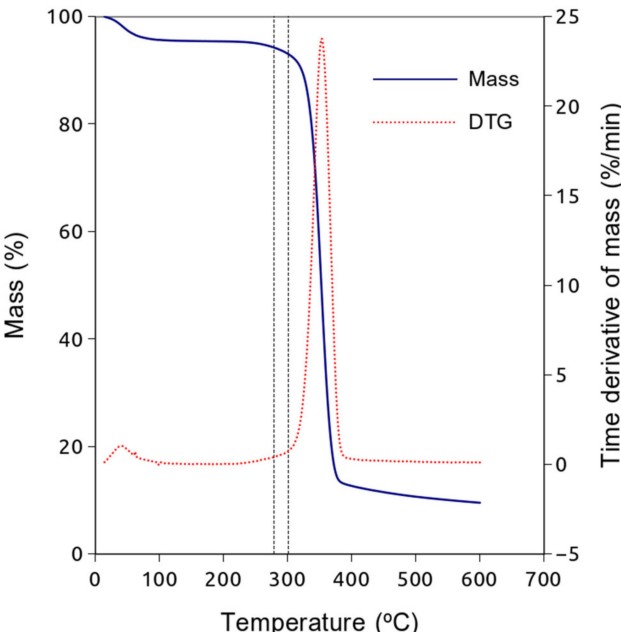

**Figure 1.** Plots of the thermogravimetry (TG) and derivative thermogravimetric (DTG) curves obtained in ramp. The range of temperature into which isothermal tests are performed is marked with vertical dashed lines.

The experimental data were analyzed through a model fitting method in two steps:

1.  Fitting of single time-derivative thermogravimetric curves obtained at different temperatures versus time. The curves are fitted to time derivative logistic functions, (Equation (1)). The fittings were performed with the Gnumeric software.
2.  Fitting and analysis of the rate parameter values obtained at different temperatures versus temperature. For this task, the R software was used, and Equations (2) and (5) were tested and compared.

Thus, a specific non-linear function is used to fit isothermal data, whereas the second step consists of identifying, describing, and modelling the strong non-linear relation between the degradation rate, estimated from the isothermal experiment fittings, and temperature. While for the first task a time derivative logistic function is used, providing accurate fittings of experimental data, for the second task an exponential function will be tested. Thus, the resulting model in this case is fully described when the exponential function is embedded into the time derivative logistic.

## 3. Results

Figures 1 and 2 show the TG and DTG curves obtained in a temperature ramp and in an isothermal experiment, respectively. For isothermal tests, a few temperatures were chosen in the range indicated in Figure 1. Theoretically, the area of the DTG peak represents the mass loss involved in that process. Figure 2 shows how the mass loss of the process is measured in the isothermal experiment. It is important to note that the first point was taken after the dehydration process but before establishing the isothermal condition.

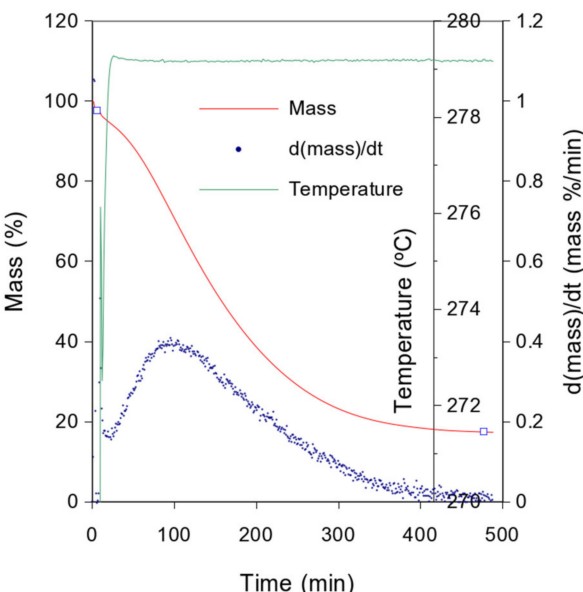

**Figure 2.** Plots of the TG and DTG curves obtained in an isothermal experiment at 279 °C. The points chosen for estimating the mass loss involved in the main degradation process are marked as squares on the mass % curve.

Figure 3 shows an isothermal DTG curve along with the temperature and the fitting of the DTG curve obtained with a derivative of a generalized logistic function. The fittings of all curves are displayed in the Appendix A Section. The parameter values resulting from the fittings are displayed, along with the average squared error (ASE) on Table 1.

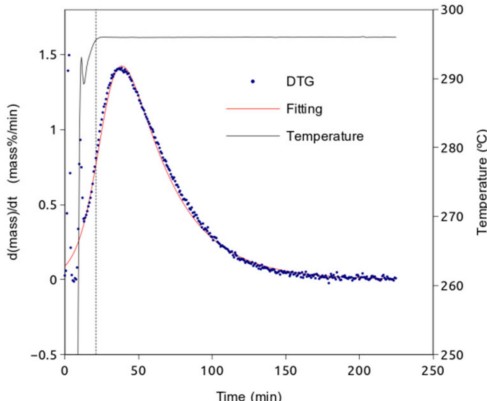

**Figure 3.** Plots of a DTG curve and its corresponding fittings. Temperature curve is also displayed. The data on the left side of the vertical dashed line were not used for fitting.

**Table 1.** Parameter values resulting from the fittings and the corresponding average squared error.

| Temperature (°C) | C (Mass %) | $\tau$ | b (min$^{-1}$) | ASE |
|---|---|---|---|---|
| 279.2 | 80.18 | 3.28 | 0.03 | $2.77 \times 10^{-4}$ |
| 284.1 | 82.55 | 3.52 | 0.05 | $3.99 \times 10^{-4}$ |
| 288.1 | 83.17 | 3.63 | 0.07 | $4.31 \times 10^{-4}$ |
| 291.1 | 81.5 | 3.78 | 0.09 | $1.08 \times 10^{-3}$ |
| 294.1 | 83.22 | 3.78 | 0.11 | $1.01 \times 10^{-3}$ |
| 296.0 | 85.62 | 3.89 | 0.12 | $6.08 \times 10^{-4}$ |
| 298.0 | 84.68 | 4.41 | 0.14 | $5.84 \times 10^{-4}$ |
| 300.9 | 85.5 | 4.43 | 0.19 | $6.13 \times 10^{-4}$ |

Figure 4 shows the plot of the rate factor, b, versus temperature. A clear exponential trend can be observed. In fact, a very good fitting was obtained with the Arrhenius expression.

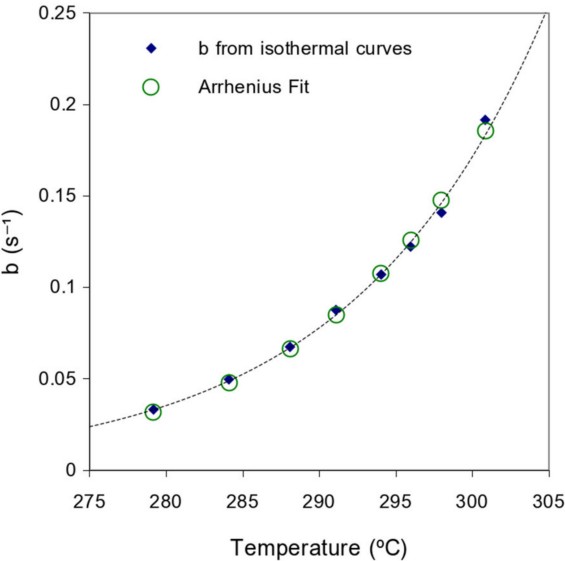

**Figure 4.** Plot of the rate parameter values obtained at different temperatures along with the fitting obtained through the Arrhenius equation.

A graphical comparison of the fittings obtained with the exponential parametric model and the non-parametric one is displayed in Figure 5.

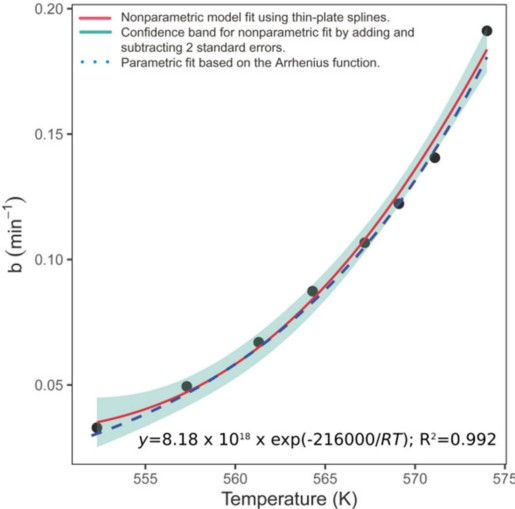

**Figure 5.** Fittings obtained with the exponential and the non-parametric models.

Figure 6 shows how the typical plot of the natural logarithm of the rate factor versus $1/T$ follows a linear trend in agreement with the Arrhenius equation. An activation energy, $E_a$, of $2.16 \times 10^5$ kJ mol$^{-1}$ was obtained along with a frequency factor, A, of $4.91 \times 10^{20}$ s$^{-1}$. In case of using the alternative expression to that of Arrhenius, a characteristic temperature $T_c = 2.60 \times 10^4$ K and a characteristic time, $t_c$, of $5.5 \times 10^{-21}$ s at that temperature are obtained.

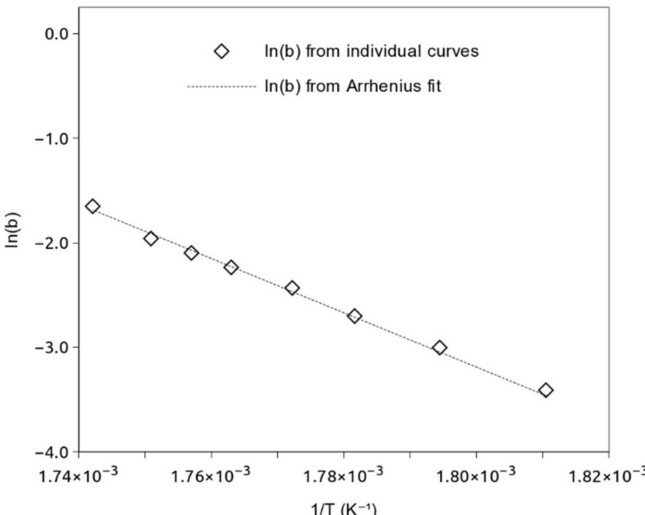

**Figure 6.** Plot of ln(*b*) versus $1/T$.

## 4. Discussion

Only isothermal data were used for fittings in this work because important differences of temperature can exist into the sample when using heating ramps, as published elsewhere [23].

Isothermal temperatures were chosen from Figure 1 with the aim of ensuring that the reaction is neither too fast nor too slow. A high temperature would imply a very fast reaction rate so an important part of the process would occur before the isothermal condition is established. Only the data recorded under the isothermal condition can be used for the fitting. On the other hand, a very low temperature would imply a very slow reaction and thus a too long experimental time.

The mass loss rate, represented by the DTG curve, is assumed to be proportional to the reaction rate. Thus, that signal is used for the kinetic analysis. As demonstrated in previous works of the authors, DTG curves can be generally fitted by time-derivative generalized logistic functions of the form

$$y(t) = [c \cdot b \cdot \exp\left(-b \cdot (t_{apm} - t)\right)] / [1 + \tau \cdot \exp(-b \cdot (t_{apm} - t))]^{\cdot (1 + \tau)/\tau} \tag{1}$$

where $t_{apm}$ is the time at the peak maximum, $c$ represents the area of the peak, $\tau$ is the symmetry factor, with $\tau = 1$ for perfect symmetry, and $b$ is a rate factor that depends on the temperature. Since the area of the peak, $c$, represents the mass loss involved in the process, the value of that parameter can be measured from the TG curves and should be similar in all samples except for differences between the samples or from their initial moisture content. In practice, that amount was directly measured on the TG curves as indicated in Figure 2. The first point was taken just after conclusion of the drying process, before establishment of the isothermal condition. The last point was taken at the end of experiment, when the mass was almost constant.

The fittings of the $b$, $\tau$, and $t_{apm}$ parameters for each individual isothermal curve was performed through the Gnumeric software using the Nlsolve algorithm implemented in that software [29].

It can be observed that the fittings presented in Figure 3 and in the Appendix A Section are very good except for the initial part of the curve, which was not included in the fitting since it corresponds to the period of time before reaching the isothermal condition. The ASE values displayed on Table 1 also confirm that visual appreciation. Being $c$ the mass involved in the whole process, a similar value is theoretically expected at any temperature, regardless if the process started to be recorded in isothermal conditions at a different extent in each case. As mentioned before, the approach used here consisted of directly measuring the mass loss from a point just after the dewatering step to the end

of the experiment, where the process was almost completed. The differences observed in the $\tau$ parameter come from little changes in the symmetry of the DTG peak. The changes are usually related little changes in sample placement.

It can be observed in Figure 4 how the *b* parameter values calculated from different tests clearly follow an exponential trend. While in crystallization processes, a Gaussian dependence of the rate factor, *b*, on temperature was found [24]; now it follows an exponential trend. That is not surprising since it is well known that crystallization and thermal degradation are intrinsically different and while the crystallization rate has a maximum somewhere in the middle of the glass transition temperature and the melting temperature, the degradation rate always increases with temperature.

One of the most used exponential expression is the Arrhenius one, which makes use of a frequency factor, A, and an activation enthalpy, $E_a$.

$$k = A \cdot \exp((-E_a)/(R \cdot T)) \tag{2}$$

where *k* represents the rate factor and R is the gas constant.

According to the determination coefficient obtained for nonlinear models, the regression model as a function of temperature explains the 99.2% of the overall information of the *b* parameter. It means that the *b* parameter can be determined exclusively based on the value of temperature.

The statistical analysis of Equation (2) with respect to the *b* parameter values obtained from the isotherms displayed in Table 1 was performed with the *DEoptim* package of the R software [30].

As a measure of goodness of fit, the determination coefficient for nonlinear models has been calculated [31]. The fitting obtained by the exponential model was compared with the regression obtained by the application of a nonparametric thin-plate spline model, which was fitted by means of the *mgcv* R software package for Generalized Additive Models [32]. The nonparametric fitting does not assume any expression that defines the relationship between the two variables. In theory, it is the most flexible model, the one that best fits the data. Therefore, the closer the fit of any parametric model to the nonparametric one, the better the quality of the parametric fit. It can be observed in Figure 5 that both fittings are almost matching along the whole range of temperatures. In addition, the parametric model is inside the confidence interval developed by adding and subtracting 2 standard errors from the nonparametric fitting (green area). Thus, we can assume that the two fittings are equivalent and, therefore, the exponential regression fit is the best of possible fittings. This fact supports the use of the exponential model based on the Arrhenius function, or any one mathematically equivalent, to define the relationship between *b* and temperature.

Figure 6 shows how the typical plot of the natural logarithm of the independently obtained rate factor values versus $1/T$ follows a linear trend in agreement with the Arrhenius equation. Using the Arrhenius expression, an activation energy, $E_a$, of $2.16 \times 10^5$ kJ mol$^{-1}$ can be obtained along with a frequency factor, A, of $4.91 \times 10^{20}$ s$^{-1}$. It deserves mention that these values cannot be compared to that obtained from other non-logistic models, even in the case of using the Arrhenius expression, too. The reason for that is that the rate parameters, regardless of whether or not they follow an Arrhenius trend, are intrinsically different if the models from which they come are different. Most Arrhenius based models can be written as

$$d\alpha/dt = k \cdot f(\alpha) \tag{3}$$

where $\alpha$ is the conversion and *k* is the rate factor, described by the Arrhenius expression, Equation (2). The model used here, described by Equation (1), can be re-written as a function of the conversion [24]:

$$d\alpha/dt = c \cdot b \cdot \exp[-b \cdot (t_{apm} - t)] \cdot [(1 - \alpha)]^{1 + \tau} \tag{4}$$

where $c$ is the amount of sample involved in the process, $b$ is the rate factor, and $t_{apm}$ the time at the peak maximum. A simple comparison of Equations (3) and (4) shows that the rate factors used in both equations are intrinsically different. While in Equation (3), the rate factor simply multiplies a function of $\alpha$, in the logistic model, Equation (4), the rate factor appears both in the exponential and the pre-exponential terms. Thus, the parameters describing that rate factors cannot be compared regardless of whether or not they follow an exponential trend.

On the other hand, the statistical analysis is equally valid if the Arrhenius expression is replaced by a mathematically equivalent one which makes use of a characteristic temperature instead of an activation energy:

$$b = \text{A} \cdot \exp((-T_c)/T) \tag{5}$$

where $T_c$ is a characteristic temperature. A characteristic time, $t_c$, can also be obtained as the inverse of $b$ at that temperature

$$t_c = \text{e}/\text{A} \tag{6}$$

A characteristic temperature, $T_c$, of $2.60 \times 10^4$ K and a characteristic time, $t_c$, of $5.5 \times 10^{-21}$ s at that temperature were obtained. Thus, the temperature dependence of the rate factor can be equally well described through Arrhenius-like parameters or through a characteristic temperature and a characteristic degradation time at that temperature.

## 5. Conclusions

A kinetic model was successfully tested on isothermal pyrolysis of cellulose at several temperatures. This model represents the degradation rate by a time derivative logistic function which includes a rate parameter that was observed to follow an exponential trend. The rate parameter values obtained by this model cannot be compared to those obtained from other intrinsically different models, even in the case of both following a formally equivalent trend. The other parameters account for the process symmetry and for the amount of sample involved in the process.

Both an Arrhenius expression and an exponential function based on a characteristic temperature are mathematically equivalent to produce a very good fitting of the rate factor obtained from the logistic fittings of individual isothermal curves.

A comparison with a previous study of crystallization kinetics from DSC data shows that, although the reaction rate has in both cases a completely different temperature dependence, the isothermal curves are still accurately described by time derivative logistic functions. Thus, leaving the rate factor analysis for a second step provides flexibility to the method and allows to extend it to different processes.

**Author Contributions:** Conceptualization, J.L.-B. and R.A.; methodology, S.N.; software, J.T.-S., A.Á.-G. and A.M.D.-D.; validation, S.N., J.T.-S. and A.M.D.-D.; formal analysis, J.L.-B.; investigation, all authors; resources, J.T.-S., S.N., J.L.-B. and R.A.; data curation, A.Á.-G.; writing—original draft preparation, R.A. and J.L.-B.; writing—review and editing, R.A., J.L.-B. and J.T.-S.; visualization, A.D.-D. and A.Á.-G.; supervision, S.N., J.T.-S. and J.L.-B.; project administration, R.A.; funding acquisition, all authors. All authors have read and agreed to the published version of the manuscript.

**Funding:** This research was funded by MINECO, grant number MTM2017-82724-R" and by Xunta de Galicia (Grupos de Referencia Competitiva ED431C-2020-14 and Centro de Investigación del Sistema universitario de Galicia ED431G 2019/01), all of them through the ERDF.

**Institutional Review Board Statement:** Not applicable.

**Informed Consent Statement:** Not applicable.

**Data Availability Statement:** Not applicable.

**Acknowledgments:** The research has been supported by MINECO grant MTM2017-82724-R, and by the Xunta de Galicia (Grupos de Referencia Competitiva ED431C-2020-14 and Centro de Investigación del Sistema universitario de Galicia ED431G 2019/01), all of them through the ERDF.

**Conflicts of Interest:** The authors declare no conflict of interest.

## Appendix A

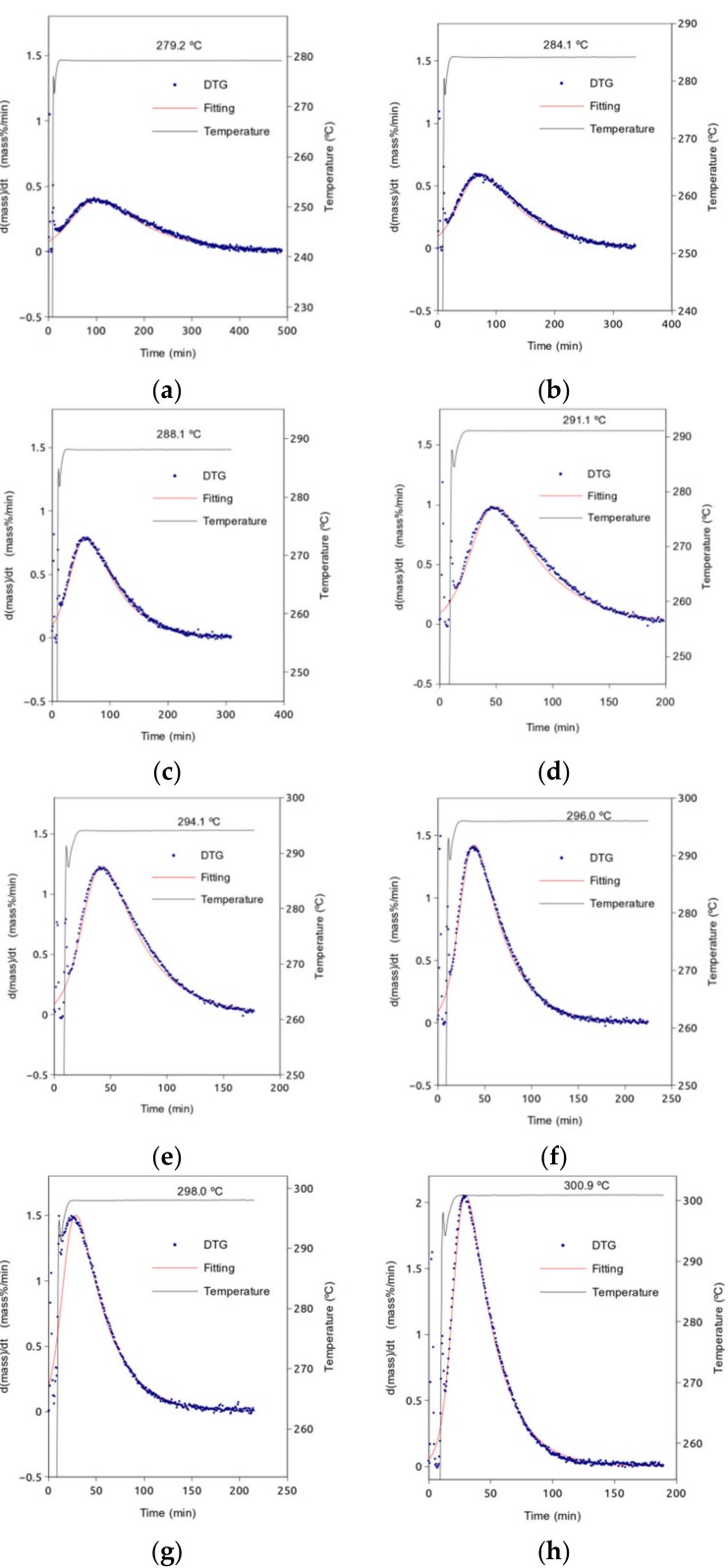

**Figure A1.** Plots of all DTG curves with their corresponding fittings at the indicated temperatures: 279.2 °C (**a**), 284.1 °C (**b**), 288.1 °C (**c**), 291.1 °C (**d**), 294.1 °C (**e**), 296.0 °C (**f**), 298.0 °C (**g**), and 300.9 °C (**h**).

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
