# Peer review of "A Logistic Approach for Kinetics of Isothermal Pyrolysis of Cellulose"

_processes, doi:10.3390/pr9030551_

Round 1

Reviewer 1 Report

The authors study the thermal behavior of cellulose. However, the material under study is not fully described and it is not easy to understand the results obtained. It is very important to know the amount of hemicellulose, lignin and other impurities or additives is contained in the sample under study. The presence of such will radically affect the thermal behavior of cellulose and chemical reactions occurring during pyrolysis (for example - Mironova, M. et al. Improvement in Carbonization Efficiency of Cellulosic Fibers Using Silylated Acetylene and Alkoxysilanes. Fibers 2019, 7, 84. https://doi.org /10.3390/fib7100084).The introduction should show the relevance of the proposed model and its place in the description of the thermal behavior of cellulose. The discussion part must be supplemented with arguments and intermediate conclusions. It is better to remove "to a previous study" in the Conclusions.

Line 11. "a" should be replaced by "at"
Line 12. Replace "the".
Line 20-21. Sentences can be combined.
Line 49. "an accurate". 

Author Response

Thanks to this reviewer for his/her comments to improve the article.

The description of the material under study was completed. It is a high purity cellulose in powder form for partition chromatography.

It is also now commented that the presence of impurities will radically affect the thermal behavior of cellulose and chemical reactions occurring during pyrolysis according to Mironova, M. (Lines 34-37).

The relevance of the proposed model and its place in the description of the thermal behavior of cellulose are now presented in the introduction section.

The Introduction (lines in the pdf document with Visible Changes 20-23, 34-37, 51-54, 64-71), Materials and Methods (lines 73-76, 98-105), and Conclusions (lines 264-275) sections were slightly modified in order to better show the potential of the method.

The following typographic/writing corrections were also applied.

Line 11. "a" should be replaced by "at"

Line 12. Replace "the".

Line 20-21. Sentences can be combined.

Line 49. "an accurate". (line 54 in Visible changes pdf)

Line 39. It

Line 40. Pyrolysis

Line 78. differential scanning calorimetry (DSC)

Line 84. A point "."

Line 109. and... experiment

Line 114. establishing

Line 131. "are" is replaced by "is".

Table 1. Format was applied to superindex (min-1)

Reviewer 2 Report

A kinetic model is proposed to fit isothermal thermogravimetric data obtained from celulose in an inert atmosphere at different temperatures. The approach lacks innovation and doesn't convince the reader for its wide use and applicability. More experimental work is needed in order to confirm the first results.

Author Response

The authors thank the reviewer comments that give us the opportunity of improving the description and scope of our work. We consider that the approach used in this work is singular and innovative: it consists of using different functions to fit isothermal data and to fit the rate parameter values resulting from the isothermal fits. A specific non-linear function is used to fit isothermal data, whereas the second step consists of identifying, describing and modelling the strong non-linear relation between the degradation rate (estimated from the isothermal experiment fittings) and temperature. While for the first task a time derivative logistic function is used (providing accurate fittings of experimental data), for the second task an exponential function was found to work perfectly. Thus the resulting model in this case is fully described when the exponential function is embedded into the time derivative logistic. As far as we know, this approach was not previously used and we did not find reports of multi-temperature fittings as good as those presented here. In addition, this work allows estimating, with relative accuracy and precision, the rate of cellulose degradation at a given temperature.

The Introduction (lines in the pdf document with Visible Changes 20-23, 34-37, 51-54, 64-71), Materials and Methods (lines 73-76, 98-105), and Conclusions (lines 264-275) sections were slightly modified in order to better show the potential of the method.

The following typographic/writing corrections were also applied.

Line 11. "a" should be replaced by "at"

Line 12. Replace "the".

Line 20-21. Sentences can be combined.

Line 49. "an accurate". (line 54 in Visible changes pdf)

Line 39. It

Line 40. Pyrolysis

Line 78. differential scanning calorimetry (DSC)

Line 84. A point "."

Line 109. and... experiment

Line 114. establishing

Line 131. "are" is replaced by "is".

Table 1. Format was applied to superindex (min-1)

We agree with the reviewer that we should try to extend applicability of the proposed method and, for that, it should be applied to other processess. While this article focuses on the application of the method to the pyrolysis of cellulose, regardless of how broad the field of application may be, we are currently working on other degradation processes. In addition, a similar approach was successfully used for PLA crystallization observed by DSC. In that case, the isothermal curves were fitted similarly to those here, but, for the rate parameter versus temperature, instead of exponential, a bell-shaped trend was found. A simple comparison of the two cases supports the current two-step approach of separating the logistic curve fitting from the subsequent analysis of the rate parameter.

Reviewer 3 Report

Cellulose is the most abundant polymer in our planet. It is used for several applications. This article describes a kinetic model for the isothermal pyrolisis of cellulose. The manuscript is well-written and the results are correctly shown. I reccomend for publication in Processes. 

Author Response

The authors strongly thank the favorable comments of Reviewer 3.

Round 2

Reviewer 1 Report

Lines 12, 13. "The first step is based on previous works of the authors and makes use of derivative of logistic functions". If the authors consider acceptable, then I would recommend that they remove "previous works of the authors".

It is also desirable to increase the font size of the labels on the axes of the graphs. 

Author Response

Thank you to Reviewer 1 for helping to improve this article.

The following changes were made according to reviewer's comments:

The sentence "The first step is based on previous works of the authors and makes use of derivative of logistic functions" was deleted.

The font size of the labels on the axes of the graphs was increased from 12 to 14.

Reviewer 2 Report

Main innovation points are now presented in the revised manuscript.

Author Response

Thank you to Reviewer 2 for the comments in the first round and for accepting the present version of the manuscript.